# Nutritional Management of Thyroiditis of Hashimoto

**DOI:** 10.3390/ijms23095144

**Published:** 2022-05-05

**Authors:** Yana Danailova, Tsvetelina Velikova, Georgi Nikolaev, Zorka Mitova, Alexander Shinkov, Hristo Gagov, Rossitza Konakchieva

**Affiliations:** 1Department of Animal and Human Physiology, Faculty of Biology, Sofia University “St. Kliment Ohridski”, 8 Dragan Tsankov Blvd., 1164 Sofia, Bulgaria; jsdanailova@uni-sofia.bg (Y.D.); hgagov@uni-sofia.bg (H.G.); 2Department of Clinical Immunology, University Hospital Lozenetsz, Sofia University St. Kliement Ohridski, 1 “Kozyak” St., 1407 Sofia, Bulgaria; 3Department of Cell and Developmental Biology, Faculty of Biology, Sofia University “St. Kliment Ohridski”, 8 Dragan Tsankov Blvd., 1164 Sofia, Bulgaria; gn_georgiev@uni-sofia.bg (G.N.); r.konakchieva@biofac.uni-sofia.bg (R.K.); 4Institute of Experimental Morphology, Pathology and Anthropology with Museum, Bulgarian Academy of Sciences, Acad. G. Bonchev St., Blvd. 25, 1113 Sofia, Bulgaria; zorkamitova@gmail.com; 5Department of Endocrinology, Medical Faculty, Medical University of Sofia, 2 Zdrave St., 1431 Sofia, Bulgaria; ashinkov@medfac.mu-sofia.bg

**Keywords:** autoimmune Hashimoto’s thyroiditis, proinflammatory, anti-inflammatory nutrients, detoxification, ecological diet

## Abstract

Since the thyroid gland is one of the organs most affected by autoimmune processes, many patients with thyroiditis of Hashimoto (TH) seek medical advice on lifestyle variance and dietary modifications to improve and maintain their hyroid function. In this review, we aim to present and discuss some challenges associated with the nutritional management of TH, focusing on environmental and dietary deficits, inflammatory and toxic nutrients, cyanotoxins, etc. We discuss the relationships among different diets, chronic inflammation, and microbiota, and their impact on the development and exacerbation of TH in detail. We share some novel insights into the role of vitamin D and melatonin for preserving thyroid function during chronic inflammation in autoimmune predisposed subjects. A comprehensive overview is provided on anti-inflammatory nutrients and ecological diets, including foods for cleansing and detoxification, which represent strategies to prevent relapses and achieve overall improvement of life quality. In conclusion, data from biomedical and clinical studies provide evidence that an appropriate dietary and lighting regimen could significantly improve the function of the thyroid gland and reduce the reactivity of autoantibodies in TH. Compliance with nutritional guidelines may help TH patients to reduce the need for medicines.

## 1. Introduction

The thyroid gland is the organ most affected by autoimmune processes [1]. Between 20% and 40% of American Caucasians and British citizens show lymphocytic infiltration in post-mortem specimens, while the highest percentage is typical for white females [2]. The intra-thyroidal lymphocytic infiltration induces chronic inflammation and autoimmune conditions, which most often results in autoimmune hypothyroidism or thyroiditis of Hashimoto (TH) [3]. TH development leads to scarring and destruction of the thyroid gland and is manifested by by a decrease of plasma free triiodothyronine (T3) and thyroxine (T4), elevated plasma levels of thyroid-stimulating hormone (TSH) and by the presence of antibodies to thyroid peroxidase (Ab-TPO) and thyroglobulin (Ab-Tg) [4]. It is generally accepted that the pathogenesis of TH, like other autoimmune diseases, represents the combination of environmental (i.e., lighting regimen, pollution, micronutrients, variety of physical and social factors), existential (lifestyle, hormonal status, diet, gut microbiota), as well as genetic factors that provoke immunological dysfunction and support the autoimmune destruction of the gland [4].

To treat the condition in the long term, patients with TH-associated hypothyroidism often require lifetime hormone replacement therapy with levothyroxine [5,6]. There is growing evidence of the existence of a thyroid–gut axis that controls many autoimmune disorders, and patients frequently report changes in their quality of life and thyroid function as a result of dietary modifications.

Genetic factors contribute to 70–80% of autoimmune thyroid diseases [7]. The major histocompatibility complex genes (HLA class I and II), thyroid-related genes, genes associated with thyroid peroxidase antibody synthesis (BACH2, TPO), and genes regulating immune response (CD40, CTLA4, PD1) are the common genetic factors [7,8].

From the environmental factors, a vast variety of nutrients play an important role in the onset and development of TH. High iodine intake, deficiencies of selenium and iron, inadequate intake of proteins, unsaturated fatty acids, and dietary fibers could favor TH [1,9,10]. Proinflammatory foods may induce dysbiosis and oxidative stress [11] that can cause intestinal inflammation and spread it towards different organs, including the thyroid gland [4,12,13]. The reduction and replacement of commensal microbiota caused by dietary supplementation significantly change the immune function and epithelial metabolism of the intestinal mucosa and the absorption of nutrients [11,14]. Drugs such as pembrolizumab, interferon-α, anti-retroviral therapy, and estrogens used for oral contraception or hormone replacement therapy are also crucial for TH [7,8]. Smoking and moderate alcohol consumption protect against TH, but quitting smoking may provoke this disease [8]. Immunomodulatory therapies and infections such as rubella, hepatitis C, and Epstein-Barr virus could also be responsible for the development of TH [8].

Cyanotoxins such as cylindrospermopsin (CYN) and microcystins, in addition to their general toxicity, increase the permeability of epithelial and model pseudo-epithelial layers of human intestines. They even possess the ability to affect the function of the gastrointestinal epithelium and other cell types, and thus induce “leaky gut” syndrome, inflammation, oxidative stress, and apoptosis [15]. Furthermore, microcystins dose-dependently reduce thyroid hormone levels, and influence deiodinase activity and transcription of genes related to thyroid hormones’ synthesis and metabolism [1]. Direct harmful effects of acute and chronic exposure to cyanotoxins on the hypothalamic–pituitary–thyroid (HPT) axis may lead to hypothyroidism [16,17].

Individual characteristics such as age, lifestyle, gender, pregnancy, and certain diseases, such as allergic rhinitis, prolactinoma, and subacute thyroiditis, may serve as an important predisposition or triggers for TH [7,8]. Current treatment of TH in hypothyroid subjects includes replacement monotherapy with levothyroxine, which greatly reduces relapses of the disease and slows down the progression of thyroid damage. However, a proportion of the patients continue to experience various symptoms and deteriorating overall quality of life. Unfortunately, there are limited data on any effective concomitant treatment other than levothyroxine, which by itself does not target the autoimmune processes related to disease severity. It is already known that the diet and lifestyle of patients with TH can play a key role in the management of disease episodes, which necessitates an in-depth study of complex external and internal factors. Intensive research shows that many dietary supplements have the potential to positively affect TH symptomatology due to their anti-inflammatory and antidepressant activity, thus improving the overall sense of well-being. Among the most attractive candidates which may be able to influence the severity of clinical symptoms and improve thyroid function are vitamins from the groups A, B, C, and D, fatty acids, antioxidants, phytochemicals, but also the indole-amine melatonin [4,10].

The interest in dietary vitamin D and melatonin is based on research findings of their physiological role as regulators of the production of inflammatory cytokines and prostaglandins. The controlled dietary supplementation of vitamin D and melatonin might represent an essential strategy for treating TH via their molecular mechanisms on the cellular level. Data suggest that appropriate nutritional protocols may help to decrease the chronic inflammation in the thyroid gland, other tissues, and organs, as well as to suppress or stop the thyroid gland degradation and thus improve patients’ quality of life [4,11].

In this review, we aim to present and discuss challenges associated with the nutritional management of TH, focusing on environmental factors and dietary deficits, inflammatory and toxic nutrients, cyanotoxins, etc. We analyze the relationships between different diets, chronic inflammation, and microbiota, and their impact on the development and exacerbation of TH in detail. We share some novel insights into the roles of vitamin D and melatonin for preserving thyroid function during chronic inflammation in autoimmune predisposed individuals.

## 2. Nutritional Factors Linked to TH Etiopathogenesis

### 2.1. Nutritional Deficits or Excess

The nutritional deficit or excess of some minerals and other nutrients plays an essential role in the etiopathogenesis of hypothyroidism and TH [18]. Iron and selenium participate in T3 (active hormone) and T4 (prohormone) formation, where iodine is a part of these molecules, and selenium is a cofactor of deiodinases that activates T4 by converting it into T3 or inactivates both T4 and T3 [10]. Zinc is important for T3 receptor activation and can influence thyroid function via other mechanisms [10,19]. Reduced intake of some nutrients, such as vitamins (A, B1, B5, B6, and C), proteins, and minerals (magnesium, sodium, potassium, phosphorus, chromium), may also provoke or support TH, and this is more evident in deficiencies for more than one of these nutrients [20]. Adequate intake of A, C, and E vitamins and group B is recommended in prophylaxis and prevention of thyroid diseases because of their antioxidative (for vitamins C and E), anti-neoplastic, and anti-goitrogenic protection (for vitamins A, D, and E), as well as regulation of the pituitary–thyroid axis, the iodine intake in the thyroid gland, and T3 signaling (for vitamin A) [4,10,20]. Inositol and its most abundant metabolite myo-inositol have a protective effect on the thyroid gland by improving TSH signaling and proinflammatory cytokine suppression [10].

Some of the nutritional deficits typical for TH are presented in Table 1.

### 2.2. Nutritional Elements Generating Intoxication

Some trace elements such as Se, Zn, and Fe participate in thyroid gland function, and their deficiency is critical for thyroid hormone homeostasis (Table 1). Others such as lead, cadmium, chromium, manganese, and fluoride are toxic for many organs and tissues, including the thyroid gland, and may provoke or support hypothyroidism when their levels in the circulation are in excess [18,23].

Table 2 summarizes the common harmful effects of toxic trace elements on thyroid hormones’ synthesis and regulation.

### 2.3. Cyanotoxins and Thyroid Function

Cyanotoxins are a diverse group of toxins produced by cyanobacteria. Their amount increases exponentially during cyanobacteria bloom in sweet or salt waters. In this case, their poisonous substances achieve high concentrations that are sufficient to harm or even kill animals and humans [36,37]. Cyanotoxins microcystins, CYN, and lipopolysaccharides were linked to gastrointestinal complaints and effects on the immune system, including gastrointestinal inflammation [38]. Microcystin significantly alters the mouse gut microbiome and induces dysbiosis [39]. Microcystins are potent and specific inhibitors of protein phosphatases 1 and 2 A and can induce oxidative stress [40]. CYN can also lead to oxidative stress, either directly or indirectly linked to the reduction of glutathione formation [39]. Furthermore, CYN may reduce the viability of human gastrointestinal epithelial cells in culture and increases the permeability of intestinal epithelium [15,41]. Cyanotoxins are also shown to facilitate the absorption of other toxins due to their inflammatory action on the gastrointestinal border [38].

Additionally, cyanotoxins may directly affect the thyroid gland. It was recently reported that microcystin-LR is able to affect the HPT, the hypothalamic–pituitary–gonadal, and the hypothalamic–pituitary–adrenal endocrine axes in rats. In regard to the HPT axis, microcystin-LR increased the concentration of TSH and decreased TRH and plasma levels of free T3 and T4 [17]. All three axes under study were influenced at the gene transcription level of the hormones and the nuclear hormone receptors. These results are in line with the effect of chronic oral administration of low doses of microcystin-LR, which leads to activation of p38/MAPK and MEK/ERK cell signaling that up-regulates type 3 deiodinase expression in mice [16].

In conclusion, drinking water and foods contaminated with cyanotoxins may indirectly influence plasma levels of free T3 and T4 via proinflammatory mucosal reaction and dysbiosis, or by a direct effect on the HPT axis and the thyroid in particular. Some of these stimuli may challenge the thyroid function and can be linked to TH. Effects of other toxins, different from the nutritional factors and cyanotoxins discussed above, on the thyroid gland and TH are not in the scope of this review.

### 2.4. Diets Favoring Inflammation and Gut Dysbiosis and Their Role in Thyroiditis of Hashimoto

Diet and microbiota are among the main factors in gut inflammation and proper intestinal function [11,13,42]. Foods rich in antioxidants help the body control oxidative stress and exert anti-inflammatory effects. These foods are considered healthy and good for maintaining an optimal body mass index. On the other hand, some diets are rich in proinflammatory foods and thus have a substantial impact on the inflammation in the human body. Research shows that certain foods can affect C-reactive protein (CRP) production, which is a serum marker of inflammation [43]. The consumption of some foods causes the release of inflammatory messengers that raise the risk of chronic inflammation, cancer, diabetes, metabolic syndrome, autoimmune diseases, and other chronic conditions (Table 3).

Accumulated data for different diets and their effects on inflammation and microbiota are summarized in Table 3.

The disruption of gut microbiota, also known as gut dysbiosis, is influenced by the individual genetic profile, diet, antibiotics, and inflammation. It is linked to the pathogenesis of some inflammatory diseases, such as obesity and inflammatory bowel disease [60]. Intestinal dysbiosis and increased intestinal permeability seem to favor the progress of TH as well, while no alteration in systemic cytokines could be detected within the same group of study [13]. Nevertheless, dietary modulation of the microbial gut balance affects the inflammatory environment, most probably due to the microbiota metabolites. Microbiota-derived metabolites, short-chain fatty acids, and Gram-negative bacterial lipopolysaccharides (LPS) exert anti-inflammatory or proinflammatory effects by acting on macrophages, depending on their M1 and M2 phenotypes [61]. Butyrate, a commensal microbial fermentation product, has been shown to favor the polarization and function of M2 macrophages through interleukin 4 (IL-4)-mediated STAT6 transcription, which could attenuate intestinal inflammation in mice [62].

A simulation model of interaction between thyroid follicular cells, Th1, Th17, Tregs, and gut microbiota in TH pathogenesis was recently proposed by Salazar-Viedma et al. [63]. Their model showed that increased proliferation and differentiation of Th1 and Th17 lymphocytes indirectly trigger inflammation and apoptosis of healthy thyrocytes. Furthermore, imbalanced gut microbiota composition results in a reduction of Treg cells and stimulation of Th17 lymphocytes, thus contributing to inflammation processes and apoptosis of healthy thyrocytes [63].

The impact of foods, the diversity of microbiota metabolites, and their interference with the immune cells’ balance in patients with TH are scarce and require further clarification to understand the mechanisms behind the development of this disease and find nutritional strategies for its alleviation.

### 2.5. Proinflammatory Nutrients Favoring Clinical Manifestation of Endocrine and Metabolic Disease

Some foods can be described as proinflammatory or anti-inflammatory depending on their content—proinflammatory or anti-inflammatory nutrients. A diet rich in too many proinflammatory nutrients may increase the risk of chronic inflammation and could accelerate the inflammatory disease process. Therefore, it is essential to not only recognize foods which can elicit inflammation, but also the inflammatory nutrients that are present in the food (Table 4).

Some foods associated with weight gain and an increased risk for chronic diseases such as heart disease and type 2 diabetes are also linked to elevated inflammatory reactions. Again, it is because some of the food ingredients or nutrients may have independent effects on inflammation.

## 3. The Role of Thyroid–Gut Axis and the Influence of Gut Microbiota on TH

There is mounting evidence for the existence of a robust thyroid–gut axis. It is reflected by a significant influence of the intrinsic bacterial microflora in the gut on the immune system reactivity and the thyroid function [71]. Furthermore, the concomitance of thyroid- and gut-related disorders is common, such as TH/Graves’ disease and celiac disease/non-celiac wheat sensitivity. Thyroid diseases are frequently associated with dysbiosis.

Dysbiosis may significantly impair the immune system and compromise inflammatory control, causing autoimmune illnesses such as autoimmune thyroid diseases [14,71]. Moreover, microbiota influences the intake of thyroid-related minerals such as iodine, selenium, zinc, and iron. They all have a role in thyroid function, and there is a definite correlation between thyroid disease and changed amounts of these minerals in the body [71].

Aside from that, it appears that Lactobacillaceae and Bifidobacterium spp. have a negative association with dietary iron and a favorable correlation with selenium and zinc. Since these bacteria are reduced in TH and Grave’s disease, it has been assumed that gut composition and mineral regulation may play a role in both disorders [14].

Furthermore, Lactobacillaceae and Bifidobacteriaceae are frequently decreased in hypo- and hyperthyroidism. Supplementation with *Lactobacillus reuteri* improved thyroid function in rats by increasing free T4, thyroid gland mass, and physiological indices, such as more dynamic behavior. This result might be caused by interleukin-10, which is known to enhance the T-regulatory cells [72].

Symbiotic supplementation is a mixture of pro- and pre-biotics that has been shown in a recent study to benefit individuals with hypothyroidism by considerably lowering TSH, levothyroxine dosage, and exhaustion, while raising fT3. However, no effect on anti-TPO or blood pressure was demonstrated [29].

It has not been determined whether bacterial infections can cause autoimmune thyroid diseases or influence therapy efficacy and prognosis [73].

Considering the numerous possible effects of microbiota and micronutrients on thyroid functions and medicines, innovative treatment methods for the management of thyroid illnesses might be developed and tailored to individuals based on their gut flora composition. Future detailed research in humans is of particular importance to delineate the influence of gut microbiota on thyroid function and disease.

Gut–thyroid interaction, and its relation with dysbiosis, changes in the immune response, increased intestinal permeability, and inflammation are presented on Figure 1.

## 4. Endocrine and Immune Regulators Involved in Adaptation: Vitamin D and Melatonin

### 4.1. Essential Effects of Vitamin D and Melatonin on Body Physiology and Cell Function

Both vitamin D and melatonin play an essential role in body physiology by controlling a variety of endocrine and immune responses. Their endogenous rhythmic production is related to the environmental cycling of light and darkness and is essentially important for ubiquitous cell physiology. Since proper circadian activity of immune and endocrine axes is a remarkable sign of overall health, vitamin D and melatonin are both promising candidates to be involved in the nutritional management of TH [74].

Data from clinical studies reveal increased prevalence of autoimmune multiple sclerosis (MS) in countries at high latitudes, where the natural lighting regimen favors vitamin D deficiency and high melatonin levels. Findings from a randomized, double-blind study on interferon beta (IFN-β)-treated patients with MS showed that melatonin secretion is negatively correlated with alterations in serum 25-hydroxyvitamin D_3_ (25-OH-D_3_), which may suggest potential interactions between both hormones with beneficial neuro-immunomodulating effects under deteriorated immune responses [74].

Vitamin D is synthesized in the skin when exposed to ultraviolet radiation from the sun and can be present in certain foods, similar to melatonin and its derivatives [75]. Both hormones have been shown to be involved in the modulation of processes such as proliferation and differentiation of normal and cancer cells, cardiovascular function, as well as immunomodulation [76]. Melatonin, known as the “message of darkness” to body physiology, is a hormone produced with a circadian pattern from the pineal gland during the dark period of the day–night cycle. Circadian rhythmicity of hormonal secretion and immune cells’ reactivity is a remarkable feature of healthy physiology which ensures adaptation to environmental stimuli by maintaining sensitive hormone–receptor interactions and regulatory negative feedback loops. The rhythmic pattern of endogenous melatonin peripheral concentration is affected under several disease conditions, particularly in psychiatric and autoimmune diseases, but also in cancer [76].

Although the synthesis of vitamin D and melatonin is in opposite phases of the day–night cycle, both hormones have been shown to be involved in modulating the activity of the immune system. Authors have suggested that while light affects melatonin synthesis, vitamin D deficiency may be related to adverse changes in circulating melatonin secretion [76]. A mechanism has been proposed by which vitamin D deficiency can cause an inflammatory response in the brain, and especially in the pineal gland, through decreased calcium absorption and gut stasis, permeability deterioration, and a corresponding increase in systemic endotoxins transferred by the intestinal microbiota.

Evidence has been provided that in addition to the protective role of vitamin D with regard to immune system alertness under different infection diseases, it may have an impact in cognitive, behavioral, and mood disorders by playing a vital role in serotonin and melatonin regulation, which are linked to mental health, especially mood regulation and sleep [77].

### 4.2. Deficits of Vitamin D and Melatonin Linked to Autoimmune Predisposition

Vitamin D deficiency has been especially demonstrated in TH patients. This may be due to the insufficiency of its natural effects on multiple immune cell-type differentiation and maturation, antigen presentation, as well as regulation of the production of cytokines and chemokines. In monocytes/macrophages, vitamin D has been shown to inhibit the expression of Toll-like receptor (TLR) 2/4 and the production of inflammatory cytokines such as IL-1, IL-6, and TNF-α. Therefore, vitamin D deficiency plays a crucial pathogenic role in autoimmune diseases [78].

Deficiency in vitamin D synthesis has been linked to autoimmune disorders associated with significant sex differences due to genetic, epigenetic, hormonal, and environmental factors [76]. It is well-established that estrogens exert dose-dependent effects on the immune response by shifting T-cell subsets to a Th1 or a Th2 profile [78]. The synthesis of estrogens and their varying concentrations during different phases related to reproductive function in women have long been associated with increased immune protection in females. In addition, estrogens have been shown to improve the function of vitamin D by promoting the expression of vitamin D receptors. These differences related to gender genetics and the associated hormonal background are an interesting direction for the development of individualized therapeutic approaches to autoimmune diseases [76].

## 5. Nutritional Treatment Strategies of Thyroiditis of Hashimoto

### 5.1. Anti-Inflammatory Nutrients 

An exponentially growing number of papers are focused on anti-inflammatory foods and their impact on health. Anti-inflammatory nutrients are those nutrients that can alter the expression of the inflammatory genes. Therefore, they are employed in different nutritional regimens to reduce inflammation and re-establish hormonal balance [79].

Furthermore, implementing these nutrients in the diet can help to maintain good nutrition while silencing inflammation. Therefore, they are considered as a novel molecular cutting-edge approach. The main anti-inflammatory nutrients are presented in Table 5.

### 5.2. Adaptive Responses Involved in the Control of Immune Reactions

Melatonin has been investigated in the therapy of several autoimmune diseases, including systemic lupus erythematosus, MS, type 1 diabetes mellitus, rheumatoid arthritis, and inflammatory bowel disease, with controversial results. However, its impact in autoimmune thyroid diseases (AITD), including Graves’ disease (GD) and TH, has been examined in rare cases. For example, a recent study investigated the possible associations of single-nucleotide polymorphisms (SNPs) of melatonin receptor type 1A (MTNR1A) and 1B (MTNR1B) with AITD in an ethnic Chinese population. The authors found only genetic variants of MTNR1A but not MTNR1B to be associated with susceptibility to GD [78].

Melatonin was shown to regulate mucosal immunity and protect against gut inflammation, at least in ulcerative colitis, a chronic inflammatory disease of the colon. Nevertheless, its role in response to LPS remains controversial [78]. The rise in LPS levels expressed by the gut microbiota increases the blood LPS through gut inflammation. LPS is recognized by LPS-binding protein (LBP) in the serum, which brings the LPS to the surface of various cells such as macrophages and endothelial cells to form a complex with CD14, a receptor molecule for LPS [88]. Providing a comprehensive review of environmental and existential factors with an impact on MS, the authors use a mechanistic approach to propose a new pathway leading to neuroinflammation and MS by including different factors such as latitude, sunlight, vitamin D, melanopsin, intestinal calcium, pineal gland, gut stasis, gut endotoxins (LPS), and CD14/TLR4 [78]. They proposed a leading role of LPS produced by gut microbiota for the etiopathogenesis of MS by generating neuroinflammation and deterioration of melatonin synthesis [78]. If such a pathway can be applicable to TH and other autoimmune disorders remains to be explored.

### 5.3. Beneficial Effects of Vitamin D and Melatonin as Antioxidants

Melatonin has several biological functions, including control of circadian and seasonal rhythms, ROS scavenging, influence on puberty, etc. However, the relationship between melatonin and the HPT axis remains controversial [89]. For example, one animal study showed that TSH was higher in the melatonin-treated group than the control group. At the same time, there was no significant difference in peripheral thyroid function. In contrast, another study demonstrated that melatonin could inhibit TSH secretion and directly inhibit the secretion of T4 [89].

In regard to thyroid gland function, research has focused mostly on a specific feature of melatonin—its potent antioxidant activity. Antioxidants remove potentially damaging reactive oxygen species (ROS) generated in the cells during their life. ROS are essential and trigger the so-called oxidative reactions, which when in excess can cause damage to macromolecules, thus violating their proper functioning [90]. In the thyroid gland, ROS are necessary for completing the synthesis of thyroid hormones. Some research showed that parafollicular cells of the thyroid gland which secrete calcitonin are capable of serotonin and melatonin production, and that this is influenced locally by TSH. It seems that melatonin and TSH balance themselves out, but there is no convincing evidence showing this. It is generally accepted that alterations in the circadian pattern of melatonin secretion are indicative of health problems and distress. Melatonin may block thyroid cell proliferation and thyroid hormone synthesis [89], and its use as a dietary supplement has to be controlled in regard to thyroid function to prevent compromised disease management.

Melatonin is widely accessible over the counter as a sleep aid. According to the National Institutes of Health, short-term use of melatonin supplements appears safe. Still, very little research exists on the long-term effects. A study from 2001 looked at the impact of melatonin on females with hypothyroidism, 36 of whom were perimenopausal, and 18 were postmenopausal [91]. The study revealed that the group taking melatonin at bed time showed significantly higher thyroid hormone levels than the placebo group after 3–6 months, and the participants experienced improvement of mood and overall alertness. The authors suggested that low melatonin levels due to aging may be connected with low levels of thyroid hormones, explaining why taking a melatonin supplement also improved TSH levels [91]. Unfortunately, large-scale trials to confirm that melatonin is safe and effective for people with hypothyroidism are still lacking.

### 5.4. Diets Aimed at Liver Detoxication and Cleansing

The nutritional management of TH includes liver detoxification and heavy metal cleansing. Promising studies report the detoxification effect of diets, often combined with an exercise plan, that additionally lead to weight loss and improved health [4]. Detoxification or detoxication is the physiological removal of toxic substances from the human body, which is carried out mainly by the liver, kidneys, intestines, lungs, lymphatic system, and the skin. However, when organs and systems are compromised, toxins and other impurities are not properly excreted, and the body is adversely affected by inflammation and the autoimmune response. The accumulation of heavy metals in the human body leads to severe injury to various organs, specifically the nervous, respiratory, endocrine, gastrointestinal, and reproductive systems [10,21]. Many foods were found to protect the human body via anti-inflammatory, antioxidative, anti-cancer, antibacterial, and other properties due to their detoxicating and cleansing effects on the liver, kidney, blood, gut, and other organs and tissues (Table 6).

### 5.5. Ecological Dietary Regimen for TH

Malnutrition risk in TH patients has not been adequately explored, and no precise conclusions may be drawn [103]. Currently, there is no convincing evidence for an effective clinical indication of dietary supplementation with vitamin D and melatonin as a complementary treatment against TH symptomatology. It is important to note that melatonin modulates the immune system by suppressing proinflammatory molecules in a dose-dependent fashion due to its physiological circadian pattern. The differences in its function can come from the dose when taken as a medication, 10 or 100 times higher than the levels produced generally in the healthy body [104]. The timing of assessment and dose–response dynamics may be essential aspects to consider when investigating the effects of vitamin D on mood and sleep due to potential vitamin D-induced moderations in serotonin and melatonin synthesis.

## 6. Perspectives for Nutritional Management of Thyroiditis of Hashimoto

Diet therapy for Hashimoto’s thyroiditis is focused on optimal nutrition and immune system modulation through an anti-inflammatory diet [4]. Since nutritional deficits are common in TH patients, according to observational and controlled investigations, nutritional management is considered essential for management of TH patients. There is evidence in the literature for selenium, potassium, iodine, copper, magnesium, zinc, iron, and vitamin A, C, D, and B deficiency in TH patients [4]. Additionally, it has been suggested that adequate protein consumption, dietary fiber, and unsaturated fatty acids, particularly those of the omega-3 family, play a beneficial role [103].

Melatonin therapy in autoimmune diseases has been studied in many animal models and in a few human clinical trials [10]. For all the conditions except rheumatoid arthritis, melatonin has been shown to have the potential to reduce the severity of symptoms. These findings indicate that melatonin treatment could be an important strategy for Hashimoto’s condition. However, convincing research on this topic is lacking, especially in light of melatonin’s complex role as an endocrine and immune modulator, but also as an active molecule at the cellular level. The most important question concerns the proper timing and dosing of melatonin, bearing in mind its circadian pattern of endogenous release and opposite phase to vitamin D and serotonin. It may be that melatonin-controlled dosing may be important to help for an overall improvement of adaptive responses, including psycho-emotional, neuroimmune, and endocrine circuits [104]. Certainly, more research is needed, in particular to better understand the delicate balance between melatonin and thyroid gland function.

## 7. Conclusions

For most TH patients, the hormone-replacement therapy with levothyroxine is indispensable. Nevertheless, an appropriate dietary regimen and ecological lifestyle can complement the standard treatment and favor remission of TH by improving the function of the thyroid gland, as well as by regulating the levels of TSH, T3, T4, Ab-TRO, and Ab-Tg. Other less significant parameters may be a repercussion of healthier body reactions and improvement of life quality, such as better sleep and alertness. Compliance with nutritional guidelines with a focus on the prevailing anti-inflammatory diet and controlled vitamin D dosing may help individual TH patients to reduce the need for medicines, slow down the course of the disease, and avoid relapses.

## Figures and Tables

**Figure 1 ijms-23-05144-f001:**
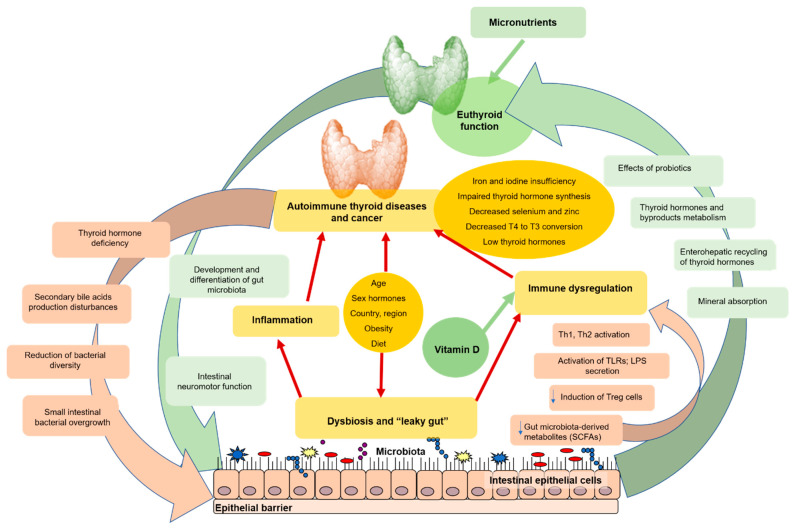
Gut–thyroid interaction in health and disease (autoimmune thyroid pathology and cancer). Thyroid diseases are frequently associated with dysbiosis. On the one hand, dysbiosis changes the immune response by encouraging inflammation and decreasing immunological tolerance, disrupting the intestinal membrane and increased intestinal permeability (a.k.a., “leaky gut”), resulting in increased antigen exposure and local inflammation. On the other hand, dysbiosis can directly affect thyroid hormone levels due to bacterial deiodinase activity and TSH inhibition [74]. The gut microbiota also regulates the absorption of thyroid-related nutrients such as iodine, selenium, zinc, and iron. All of them are required for thyroid function, and there is a definite correlation between thyroid dysfunction and changes in these minerals’ levels. Probiotics have been demonstrated to be effective in thyroid problems and can have a good effect. The healthy or diseased thyroid gland can also influence microbiota via many mechanisms, including melatonin. Legend: green arrows (wide and thin) denote a predominantly positive impact, while pink arrows (wide and narrow) indicate a primarily negative effect.

**Table 1 ijms-23-05144-t001:** Deficits or excess of nutrients and their effects on thyroid gland function.

Nutrients	Foods	Effects	Molecular Targets	Restrictions	Ref.
Iodine/I	Iodized salt, plum, maize; sea fish, iodized milk, dairy products, chicken eggs	Deficiency of iodine may cause goiter and hypothyroidism. Thyroid peroxidase in the presence of peroxide iodinates tyrosine bound to thyroglobulin	Part of T3 and T4	Chronic high dietary iodine intake may induce autoimmune thyroiditis	[1,9]
Iron/Fe	Meat, animal offal, pumpkin seeds, cocoa and bitter chocolate, sardines, seafood	Necessary for the thyroid peroxidase reaction	Thyroid peroxidase	Chronic high dietary iron has pro-oxidant and cancerogenic effects	[4]
Selenium/Se	Brazilian walnut, meat, liver, fish, spinach	Activation or inactivation of T4, antioxidant and anti-inflammatory effects	Thyroid function depends on glutathione peroxidases, deiodinases, and selenoprotein S	High doses of Se are toxic	[4,9]
Zinc/Zn	Whole grain cereals, flax seeds, pumpkin seeds, millet, meat, buckwheat	Zn deficiency leads to disturbances in T3 and T4 levels and increases antibody titers against thyroid antigens	Cofactor for over 300 metalloproteins activates deiodinases and is needed for proper T3 receptor signaling, stimulates the synthesis of TBS	Low doses of Zn cause oxidative stress	[17,19]
Vitamin D	Fish oil, fatty fish, chicken eggs	Malnutrition of vitamin D correlates to autoimmune diseases	Nuclear vitamin D receptor via vitamin D response element regulates more than 200 human genes	Hypervitaminosis leads to hypercalcemia	[4,21]
Proteins	From unprocessed meat, eggs, sea fish	Low-protein content, soy proteins diet, and starvation downregulate HPT axis; malnutrition leads to thyroid gland damage, especially in children	Low-protein diet increases plasma TBG and decreases plasma transthyretin, T3, and pituitary TSH transcript in rats, and increases TSH in humans	N/A	[20,22]
Inositol (vitamin B8)	Citrus fruit, cantaloupe, bananas, raisins, and fiber-rich foods	TSH signaling, thyroid cells’ protection; immunomodulatory effect, decreases TSH to normal values in patients with hypothyroidism when applied with selenium	In TSH signaling as part of PIP-3; T3 decreases TPO-Ab, and Tg-Ab increases thyrocytes viability in the presence of H_2_O_2_ and cytokines		[10]

Abbreviations: TBG—thyroxine-binding globulin; TBS—thyroxine-binding protein.

**Table 2 ijms-23-05144-t002:** Toxic nutritional elements.

Nutrients	Foods	Effects	Molecular Targets	Restrictions	Ref.
Manganese/Mn	Whole grains, clams, oysters, mussels, nuts, soybeans, leafy vegetables and legumes, rice, coffee, tea, black pepper, and other spices	Mn modulates TSH secretion by a dopaminergic mechanism	Part of Mn-SOD	In high doses, pro-antioxidative effect	[24,25]
Fluoride/F	Drinking water, fluoride-treated salt	Higher fluoride in drinking water increases hypothyroidism by about twice, iodine deficiency	T3 and T4 iodination	Toxic	[18]
Lead/Pb	Polluted air	Increased plasma levels in hypothyroidism, pro-oxidative effects; decreases Se in blood	Thyroid selenoproteins	Toxic	[26]
Cadmium/Cd	Smelters, food, burning fossil fuels, plastics, and nickel-cadmium batteries, cigarette smoke, phosphate fertilizers	Oxidative stress and mitochondrial, leading to hypothyroidism and hyperthyroidism; selenium and myo-inositol protect against Cd	MCP-1 and C-X-C motif chemokine 10 expression	Toxic	[18,27,28]
Chromium/Cr	High doses from air, foods, or through the skin	Via pleiotropic mechanisms, most of them indirect via insulin, cortisol, Fe, and Se	Oxidation of proteins influences Fe and Se homeostasis	Oxidative changes of proteins	[29]
Iron/Fe	Red meat, liver, beans, edamame beans, chickpeas, nuts, dried fruit (apricots), cereals, soybean flour	Facilitates thyroid hormone iodination	Activates thyroid peroxidase	In high doses, pro-antioxidant	[23]
Aluminum/Al	Through food, through breathing, and by skin contact	The aluminum ion (Al^3+^) is harmful. The uptake of aluminum can occur through food, breathing, and skin contact. Long-lasting uptakes of significant aluminum concentrations can damage the central nervous system, leading to dementia, loss of memory, listlessness, severe trembling	AlF_4_^-^ is a non-specific G-protein activator	In high doses, neurotoxin	[30]
Nickel/Ni	Hazelnuts; cocoa and dark chocolate; fruits (almonds, dates, figs, pineapple, plums, raspberries); grains (bran, buckwheat, millet, whole grain bread, oats, brown rice, sesame seeds, sunflower seeds); seafood (shrimps, mussels, oysters, crab, salmon); vegetables (beans, savoy cabbage, leeks, lettuce, lentils, peas, spinach, cabbage); tea from drink dispensers; soya and soya products; peanuts; licorice; baking powder	Contact dermatitis; headaches; gastrointestinal manifestations; respiratory manifestations; lung fibrosis; cardiovascular diseases; lung cancer; nasal cancer; epigenetic effects	Immunotoxic and carcinogenic agent	N/A	[31]
Tin/Sn	Tin is present in the air, water, soil, and landfills; it is a normal part of many plants and animals; tin concentrations in foods not packaged in metal cans are minimal; people can be exposed to the tin when consuming food or liquid from tin-lined cans.	Inhalation, oral, or dermal exposure to some organotin compounds has been shown to cause harmful effects in human skin and eye irritation, respiratory irritation, gastrointestinal effects, and neurological problems in humans exposed for a short period to high amounts of organotin compounds.	N/A	Lethal intoxication cases may appear with large amounts but are rare	[32]
Gallium/Ga	Found in small amounts in nature and the human body	Acute exposure to gallium (III) chloride can cause throat irritation, difficulty breathing, and chest pain. Its fumes can cause pulmonary edema and partial paralysis.	N/A	N/A	[33]
Genistein	Soy and soy foods contain this phytoestrogen belonging to isoflavones	Goitrogenic effect, hypothyroidism	Inhibitor of thyroid peroxidase and sulfotransferase enzymes	Long-term consumption presents a risk for infants and women	[34,35]

Abbreviations: MCP-1—monocyte chemoattractant protein-1; Mn-SOD—manganese superoxide dismutase.

**Table 3 ijms-23-05144-t003:** Diets, inflammation, and microbiota.

Diet	Foods	Eliminated Foods or Nutrients	Effects	Restrictions or Remarks	Ref.
Low-residue diet	Elimination of fruits, vegetables, whole grains, and legumes	Reduced fiber intake	Relieve obstructive symptoms; no effect on inflammation	Long-term use could decrease microbial diversity	[11,44,45]
Anti-inflammatory diet	More prebiotic and probiotic foods, n-3 PUFA, wild fish, grass-fed meat, vegetables, fruit, nuts, some saturated fat	Restriction of gluten, lactose, total fat, refined carbohydrates, others	Decrease of HBI or MTLWSI	N/A	[46]
Autoimmune diet (modified-Paleo diet)	Meat (grass-fed), fish, vegetables, excluding nightshade vegetables, sweet potatoes, fruit (in small amounts), coconut milk, avocado, olive, coconut oil, dairy-free fermented foods (kefir, kombucha, sauerkraut, kimchi)	Less or elimination of processed food, dairy, grains, refined sugars, legumes, and cereals	Remission of gut inflammation	N/A	[47]
Leaky gut diet	Glutamine, N-acetyl-L-cysteine, and zinc	Gluten- and milk-free, low-carb, and low-sugar diet	Reduces gut-derived inflammation; ROS and inflammatory cytokines IL-1β, IL-6, IFN-γ, TNF-α; prevents bacterial translocation via enhancing weakened tight junctions	N/A	[48,49]
Mediterranean diet	Fish, lean meat, whole grains, legumes, nuts, dairy, olive oil, vegetables, fruits, and moderate wine consumption	Heavily processed foods, i.e., processed red meats, refined grains, refined/processed/hydrogenated oils, alcohol, butter	Lowers the inflammatory load and simultaneously balances gut microbiota	Diet is rich in omega-3 and omega-9 FA, fibers, complex carbohydrates, minerals, vitamins, and secondary plant metabolites	[50]
Nordic diet	High consumption of whole grains and unrefined sugars, fish and lean meat, dairy products, canola or rapeseed oil, and vegetables and fruits like cabbage and berries	Refined carbohydrates and processed foods	Lower blood pressure, normalize cholesterol, lose weight, or maintain a healthy weight.	Causing obesity	[51]
Proteins (dietary)	Latex hevein, kamut, soy sauce, gelatin, scallops, cashew, Brazil nut	Reduction of other macronutrients	Cross-reactivity of dietary proteins with monoclonal antibodies against T4, T3, and Tg	The observed immunoreactivity of purified dietary proteins in vitro might have an antithyroid effect in vivo only in the very leaky intestine	[52]
Proteins (microbial)	Protein fermented food	N/A	Gastrointestinal pathogen *Yersinia enterocolitica* have proteins (porins) that mimic thyroid antigens and could lead to autoimmunization and stimulate precursor B cells for TSHR-Ab production	N/A	[53,54]
Fatigue reduction and anti-inflammatory diet	Foods rich in antioxidative vitamins, omega-3 fatty acids, and in fibers, polyphenol-rich vegetables	Reduced inflammatory foods	Anti-inflammatory effect, fatigue reduction	N/A	[55]
Western diets	Rich in linoleic acid; high ratio of ω-6 to ω-3 FA	Diet is rich in calories	Inflammatory effect	N/A	[56]
Wellnessup diet	Organic plant-based diet including various vegetables, fruits, whole grains, nuts, and phytonutrients	Elimination of meat, eggs, fish, dairy products, processed food, refined sugars	It may have several beneficial effects, such as body fat reduction and improving some of the detoxification elements through caloric restriction.	N/A	[57]
FODMAP diet	Proteins: beef, chicken, eggs, fish, lamb, pork, prawns, tempeh, and tofu; whole grains starches: white and brown rice, lentils, corn, oats, quinoa, cassava, and potatoes fruits: blueberries, raspberries, pineapple, honeydew melon, cantaloupe, kiwi, limes, guava, starfruit, grapes vegetables: bean sprouts, bell peppers, radishes, bok choy, carrots, celery, eggplant, tomatoes, spinach, cucumber, pumpkin, zucchini nuts: almonds, macadamia nuts, peanuts, pecans, pine nuts, and walnuts seeds: pumpkin, sesame, sunflower seeds, linseeds, dairy, coconut and olive oils, peppermint tea	Oligosaccharides: wheat, rye, nuts, legumes, garlic, artichokes, onion disaccharides: lactose-containing products—milk, ice cream, yogurt, soft cheese, buttermilk, condensed milk, whipped cream monosaccharides: fructose-containing fruits—apples, pears, mango, watermelon, sweeteners—honey, agave nectar, and high-fructose corn syrup polyols: mannitol and sorbitol in apples, pears, cauliflower, stone fruits, mushrooms, and snow peas; xylitol and isomalt in sweeteners, such as those in sugar-free gum and mints	Reduces symptoms of irritable bowel syndrome. May decrease both stomach pain and bloating. Helps manage flatulence, diarrhea, and constipation	N/A	[58]
Elimination diet	Common foods in the elimination diet are gluten, dairy products, citrus, soy, peanuts, eggs, corn, tree nuts, beef, refined sugars	Eliminates certain food or group of foods believed to cause an adverse food reaction, often referred to as a “food intolerance.”	May reduce inflammation and allergy symptoms	N/A	[58]
Vegan diet	Plant-based foods: fruits, vegetables, soy, legumes, nuts and nut butter, sprouted or fermented plant foods, plant-based dairy alternatives, and whole grains	Excludes meat, poultry, eggs, dairy, and seafood	Associated with improved glycemic control, lower total cholesterol, blood pressure, and BMI. May prevent cancer.	Possible deficiency of some essential amino acids and vitamins	[59]

Abbreviations: HBI—Harvey Bradshaw Index; IFN-γ—Interferon gamma; MTLWSI—Modified Truelove and Witts Severity Index; N/A—not applicable.

**Table 4 ijms-23-05144-t004:** Inflammatory nutrients.

Nutrients	Foods	Effects	Molecular Targets	Ref.
Inflammatory proteins	Avocado, orange juice, cooked Brussels sprouts, seaweed, radish, almond, Brazil nut, macadamia nut, mustard seeds, cashew, and others	Immunological reactivity of several dietary proteins to T3-Ab and T4-Ab, one with Tg and 5’ deiodinase. This could be observed in vivo only in case of a combination of improper protein digestion and leaky intestine	Specific T3, T4, Tg, and 5’ deiodinase Ag-Ab immune cross-reactivity for few dietary proteins with similarity to T3 or T4; no binding to TSH-R, TPO, or TBP; inflammatory effect only in impaired digestion and leaky intestine	[52]
Gluten	Wheat, rye, barley, and contaminated oat products	Antigen in CD; molecular mimicry with thyroid Tg; effect on gut microbiota and permeability; element deficiency due to CD; gluten-free diet for TH in the absence of CD is not recommended	Anti-Tg Ab can be produced in the comorbidity of CD and TH	[4]
Sugar: sucrose, fructose, glucose, high-fructose corn syrup (HFCS)	Refined sugar, candy, chocolate, cookies, sweets, cakes, foods with added sugar, sugary and soft drinks	Increase inflammatory markers and circulating uric acid; associated with metabolic disorders		[64]
Artificial trans fatty acids (FA)	Hydrogenated vegetable oils, margarine	Increases production of inflammatory cytokines associated with atherosclerosis; increases the concentration of plasma biomarkers of inflammation and endothelial dysfunction	Trans FA increase plasma concentrations of CRP, IL-6, soluble TNF receptor 2, E-selectin, and sVCAM-1	[65]
Refined carbohydrates	White bread, pasta, candy, pastries, cookies, cakes, sugary soft drinks, and all processed foods with added sugar or flour	Enhance the growth of inflammatory gut bacteria that can increase the risk of obesity and inflammatory bowel disease		[66]
Advanced glycation end products	Red and processed meat such as grilled meat, sausage, bacon, ham, smoked meat, and beef jerky	Increased oxidative load and higher inflammatory response	Increased N-nitrosation and oxidative load leading to DNA adducts and lipid peroxidation in the intestinal epithelium, proliferative stimulation of the epithelium, higher inflammatory response, which may trigger pro-malignant processes	[67]
Ethanol	Alcoholic beverages such as wine, beer, spirits, liquors	Increased levels of C-reactive protein (CRP) may cause a “leaky gut” condition	N/A	[68]
Meat	Meat, both processed and unprocessed	Higher meat consumption, particularly of processed meat, was positively associated with the inflammatory marker—serum CRP (mg/L)	CRP, white blood cell count, interleukin 6 (IL-6), and TNF-α	[69,70]

Abbreviations: CRP—C-reactive protein; CD—celiac disease; IL-6—interleukin 6; sVCAM-1—soluble vascular cell adhesion molecule 1; TNF-α—tumor necrosis factor-alpha.

**Table 5 ijms-23-05144-t005:** Anti-inflammatory nutrients.

Nutrients	Foods	Effects	Molecular Targets	Restrictions	Ref.
Magnesium/Mg	Yogurt, kefir, pumpkin seeds, cocoa, nuts, bitter chocolate, whole-grain cereal products, avocado, some fatty fish, green vegetables	Participates in many enzyme reactions, in melatonin synthesis, anti-inflammatory, deficit leads to immune disorders	Mg decreases C-reactive protein and antibodies against thyroglobulin (TG), part of Mg-SOD	N/A	[56,60]
Dietary fiber	Whole grains, beans, lentils, broccoli, apple, berries, dried fruits, avocado, popcorn	Support a proper gut microbiome, absorb toxins, anti-inflammatory; slows down and impedes nutrient uptake	N/A	N/A	[4]
Fatty acid	Fish, oils (flaxseed, extra-virgin olive oil, canola oil, sunflower oil), chia seeds, walnuts, avocados	Excess of saturated and deficiency of unsaturated fatty acids; provoke leaky intestinal syndrome and worse intestinal microbiota; induce cellular stress	Balance of dietary fatty acids improves gut microbiota, metabolism, and has an anti-inflammatory effect; saturated lipids induce inflammation via Toll-like receptor activation; n-3 PUFA decrease endotoxin permeability	N/A	[4,56,80]
Resveratrol	Vegetables and fruits, such as grapes, berries, and peanuts	Antioxidant, anti-inflammatory, and antiproliferative agent; short-term activation and long-term inhibition of expression and function of Na/I symporter	Na/I symporter	High doses of resveratrol will induce hypothyroidism	[81]
Gingerol, shogaol and paradols	Ginger (Zingiber officinale) roots	Phenolic substances with antioxidant and pleiotropic anti-inflammatory mechanisms regulating NO, PI3K/Akt, TNF-alpha signaling; antimicrobial and anti-cancer activity	Nrf2 enhances the expression of antioxidative enzymes, anti-inflammatory effects, glutathione, and stress-induced enzymes such as heme-oxygenase 1	N/A	[82]
Omega-9 mono- unsaturated oleic FA	Olive and fish oil	Balancing gut microbiota and inflammatory response; omega-9 mono-unsaturated oleic FA, which can be converted to anti-inflammatory eicosatrienoic acid	Inhibits leukotriene B4 synthesis	N/A	[83]
Omega-3	fish (salmon, mackerel, tuna, herring, and sardines), seafood, nuts and seeds (flaxseed, chia seeds, and walnuts), plant oils (flaxseed oil, soybean oil, and canola oil)	Regulates food intake, substrates for pro-resolving lipid mediators, anti-inflammatory effect	N/A	N/A	[56]
Sugars	candies, cakes, cookies, pies, sweet rolls, pastries, doughnuts, ice cream, sweetened yogurt, sweetened drinks, juice	Dysbiosis of gut microbiota increases gut inflammation, weight gain, and alters vagal gut–brain communication	N/A	Restrictions are needed to prevent from cardiovascular, metabolic, and other chronic diseases	[49,84]
Flavonoid and non-flavonoid dietary polyphenols	Plant sources—vegetables, fruits, dry legumes, cereals, olives, cocoa, coffee, tea, wine	Influence dendritic cells, have an immunomodulatory effect on macrophages, increase proliferation of B cells and T cells, and suppress Th1, Th2, Th17, and Th9 cells. Polyphenols reduce inflammation by suppressing the proinflammatory cytokines in inflammatory bowel disease by inducing Treg cells in the intestine, inhibiting TNF-α, inducing apoptosis, decreasing DNA damage. Role in prevention/treatment of autoimmune diseases	Downregulation of proinflammatory cytokines IL-1, IL-6, and INF-γ	N/A	[85]
Flavonoids	Plant sources—fruits, tea, vegetables, legumes, dark chocolate	Anti-inflammatory and antioxidant effect by free-radical scavengers and inhibition of their formation	Inhibition of protein kinases, phosphodiesterases, PLA_2_, COX, LOX, Modulation of NF-κB, GATA-3, and STAT-6 signaling	N/A	[85,86]
MCP	Dietary pectin is modified by pH and heat-controlled enzymatic treatment	MCP is absorbed in circulation; has anti-cancer effects; increases urinary excretion of lead, arsenic, and cadmium; reduces fibrosis in the liver, kidney, and adipose tissue; improves immune function	Antagonism of galectin-3	N/A	[87]

Abbreviations: MCP—modified citrus pectin; Mg-SOD—Mg-superoxide dismutase; PUFA—polyunsaturated fatty acids; Th—T helper; N/A—not applicable.

**Table 6 ijms-23-05144-t006:** Foods for cleansing and detoxication.

Diet	Active Ingredients	Effects	Restrictions	Ref.
Beetroot (*Beta vulgaris*)	High levels of antioxidants betalain pigments and betaine	Amplify specific enzymes that support the liver and its detoxification, choleretic effect	Beet can make urine or stools appear pink or red. But this is not harmful. There is concern that beets might cause low calcium levels and kidney damage, but this has not been shown in people.	[92]
Broccoli (*Brassica oleracea*)	Diindolylmethane, glucoraphanin	Antiviral, anti-cancer	Gas or bowel irritation caused by broccoli’s high amounts of fiber.	[93]
Lemon (*Citrus limon*)	Naringin, citric acid	Decreasing liver damage, prevents oxidative (stress-related) damage. Regulate blood pressure.	Skin irritation is the most common side effect of using fruit acids. Lemon is extremely acidic, which can irritate the skin.	[94]
Pomegranate (*Punica granatum*)	Punicalagins (pomegranate ellagitannins)	Anti-cancer	Some people have experienced sensitivity to pomegranate extract; they include itching, swelling, runny nose, difficulty breathing	[95]
Watermelon (*Citrullus lanatus*)	Citrulline, lycopene	Antitoxic, hypoglycemic	Excessive use can lead to diarrhea and other digestive troubles.	[93]
Avocado (*Persea americana*)	Rich in fatty acids; vitamins; minerals, fiber, phytochemicals	Protect against damage caused by liver toxin d-galactosamine, contains antioxidants, anti-cancer, and anti-inflammatory components	Excess use may lead to excess calories intake.	[96,97]
Apple (*Malus domestica*)	Soluble fiber pectin	Helps purge toxins from the bloodstream and lower LDL cholesterol; aid the excretion of mercury and lead	No side effects of apple fruit or apple juice; apple seeds contain cyanide and are poisonous	[98]
Turmeric (*Curcuma longa*)	Curcumin	Antibacterial, antiviral, anti-inflammatory, antitumor, antioxidant, antiseptic, cardioprotective, hepatoprotective, nephroprotective, radioprotective, and digestive activities	Turmeric usually does not cause serious side effects. Some people can experience mild side effects such as stomach upset, nausea, dizziness, or diarrhea. These side effects are more common at higher doses.	[99]
Blueberries (*Vaccinium angustifolium*)	Antioxidants (anthocyanins)	Lower blood pressure, boost vascular health, protect lungs, anti-cancer effect, prevent Alzheimer’s disease.	Blueberry fruit is safe when consumed in normal amounts	[100]
Cilantro (*Coriandrum sativum*)	Metal-binding proteins	Enhance mercury excretion and decrease lead absorption.	Rarely people might experience allergies after eating cilantro, such as hives, facial or throat swelling	[101]
Ginger (*Zingiber* *officinale*)	Phytochemicals	Boost the body’s ability to process food and eliminate waste; protects against oxidative stress, anti-inflammatory, and anti-cancer effects	Higher doses of 5 g can cause mild side effects, including heartburn, diarrhea, burping, and general stomach discomfort	[102]

## Data Availability

No new data were created or analyzed in this study. Data sharing is not applicable to this article.

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
