# Peer review of "Nutritional Management of Thyroiditis of Hashimoto"

_ijms, 2022, doi:10.3390/ijms23095144_

Round 1

Reviewer 1 Report

This comprehensive review explored current knowledge on the role of diet, namely micro- and macronutrients, and specific molecules, namely vitamin D and melatonin, in the potential management of thyroiditis of Hashimoto. The manuscript is clear and generally well written. I only point out minor comments as listed below.

Lines 51-52. Patients with TH are treated only in case of TH-associated hypothyroidism. The authors should precise this point.

Line 122. Consider adding Vitamin A as well, as mentioned later in this sentence.

Line 193. Please provide the full name of IL-4. From here on, please use the acronym only.

Really, Table 1 reports the effects on the thyroid gland related to both the deficiency and the excess of micronutrients, therefore please change the text and the title of accordingly. Furthermore, a list of abbreviations below rather within the table is suggested.

A list of abbreviations should be added below each table.

Lines 274, 275. Please provide the full name of IFN-β and 25-OH-D.

Line 307. Did the authors mean “vitamin D deficiency”?

Line 333. For multiple sclerosis, please use the abbreviation given in a previous paragraph.

Lines 343,344. Please amend the typo in “expressed”.

Line 348. Please correct the typo in “impact”.

Line 351. Consider adding “a” before “leading”.

Line 353, 354. Please use the acronym “TH”.

Line 358. Please use the abbreviation “HPT”.

Line 359. Please use the acronym “TSH”.

Line 382. Please correct the typo in “experienced”.

Line 389. Please correct the typo in “promising”.

Line 428. Please correct the typo in “active”.

Line 429. Please correct the typo in “proper”.

Author Response

Dear Sirs,

Thank you very much for your time to review once again our paper entitled “Nutritional management of thyroiditis of Hashimoto” submitted for consideration for publishing in mdpi journal International Journal of Molecular Sciences/Special edition in response to an invitation.

We have provided a POINT-by-POINT response to the reviewer’s comments here, and revised our manuscript with all changes made visible by track changes.

This comprehensive review explored current knowledge on the role of diet, namely micro- and macronutrients, and specific molecules, namely vitamin D and melatonin, in the potential management of thyroiditis of Hashimoto. The manuscript is clear and generally well written. I only point out minor comments as listed below.

  • Thank you very much for the overall evaluation of our paper as good.

Lines 51-52. Patients with TH are treated only in case of TH-associated hypothyroidism. The authors should precise this point.

  • Thank you for the valuable comment. We agree completely and corrected the mentioned issue.

Line 122. Consider adding Vitamin A as well, as mentioned later in this sentence.

Line 193. Please provide the full name of IL-4. From here on, please use the acronym only.

  • All the mentioned issues have been corrected.

Really, Table 1 reports the effects on the thyroid gland related to both the deficiency and the excess of micronutrients, therefore please change the text and the title of accordingly. Furthermore, a list of abbreviations below rather within the table is suggested.

A list of abbreviations should be added below each table.

  • We completely agree with the referees` points. We have revised the tables.

Lines 274, 275. Please provide the full name of IFN-β and 25-OH-D.

  • The issue has been corrected.

Line 307. Did the authors mean “vitamin D deficiency”?

  • Thank you for the note. In line 307 we did mean vitamin D, but the second part of the sentence regarding autoimmune diseases is associated with vitamin D deficiency. We have corrected the issue.

Line 333. For multiple sclerosis, please use the abbreviation given in a previous paragraph.

Lines 343,344. Please amend the typo in “expressed”.

Line 348. Please correct the typo in “impact”.

Line 351. Consider adding “a” before “leading”.

Line 353, 354. Please use the acronym “TH”.

Line 358. Please use the abbreviation “HPT”.

Line 359. Please use the acronym “TSH”.

Line 382. Please correct the typo in “experienced”.

Line 389. Please correct the typo in “promising”.

Line 428. Please correct the typo in “active”.

Line 429. Please correct the typo in “proper”.

  • Thank you for noticing these issues. All of them have been corrected.

Reviewer 2 Report

In the present paper “Nutritional management of thyroiditis of Hashimoto”, Yana Danailova and colleagues aimed to present and discuss challenges associated with the nutritional management of thyroiditis of Hashimoto (TH), focusing on environmental and dietary deficits, inflammatory and toxic nutrients, cyanotoxins etc. Indeed, they concluded that data from biomedical and clinical studies provide evidence, that appropriate dietary and lighting regimen could significantly improve the function of the thyroid gland and reduce the reactivity of autoantibodies in TH. Moreover, compliance with nutritional guidelines may help TH patients to reduce the need for medicines.

Overall, I think that the manuscript is intriguing, well-written (within the scope of this journal), well-structured and the data are of relevance on a current topic of interest. I would like to congratulate the authors on their work.

I have some suggestions to improve the quality of review.

Please insert for your convenience the following references with careful discussion regarding the mechanism of action of soy isoflavone genistein on thyroid (1,2) as well as of different nutraceuticals on cadmium-induced thyroid toxicity (3):

1) Marini H, Polito F, Adamo EB, Bitto A, Squadrito F, Benvenga S. Update on genistein and thyroid: an overall message of safety. Front Endocrinol (Lausanne). 2012 Jul 31; 3:94. doi: 10.3389/fendo.2012.00094. PMID: 23060856; PMCID: PMC3459182.

2) Bitto A, Polito F, Atteritano M, Altavilla D, Mazzaferro S, Marini H, Adamo EB, D'Anna R, Granese R, Corrado F, Russo S, Minutoli L, Squadrito F. Genistein aglycone does not affect thyroid function: results from a three-year, randomized, double-blind, placebo-controlled trial. J Clin Endocrinol Metab. 2010 Jun;95(6):3067-72. doi: 10.1210/jc.2009-2779. Epub 2010 Mar 31. PMID: 20357174.

3) Benvenga S, Marini HR, Micali A, Freni J, Pallio G, Irrera N, Squadrito F, Altavilla D, Antonelli A, Ferrari SM, Fallahi P, Puzzolo D, Minutoli L. Protective Effects of Myo-Inositol and Selenium on Cadmium-Induced Thyroid Toxicity in Mice. Nutrients. 2020 Apr 26;12(5):1222. doi: 10.3390/nu12051222. PMID: 32357526; PMCID: PMC7282027.

Author Response

Dear Sirs,

Thank you very much for your time to review once again our paper entitled “Nutritional management of thyroiditis of Hashimoto” submitted for consideration for publishing in mdpi journal International Journal of Molecular Sciences/Special edition in response to an invitation.

We have provided a POINT-by-POINT response to the reviewer’s comments here, and revised our manuscript with all changes made visible by track changes.

In the present paper “Nutritional management of thyroiditis of Hashimoto”, Yana Danailova and colleagues aimed to present and discuss challenges associated with the nutritional management of thyroiditis of Hashimoto (TH), focusing on environmental and dietary deficits, inflammatory and toxic nutrients, cyanotoxins etc. Indeed, they concluded that data from biomedical and clinical studies provide evidence, that appropriate dietary and lighting regimen could significantly improve the function of the thyroid gland and reduce the reactivity of autoantibodies in TH. Moreover, compliance with nutritional guidelines may help TH patients to reduce the need for medicines.

Overall, I think that the manuscript is intriguing, well-written (within the scope of this journal), well-structured and the data are of relevance on a current topic of interest. I would like to congratulate the authors on their work.

  • Thank you very much for the overall evaluation of our paper as good.

I have some suggestions to improve the quality of review.

Please insert for your convenience the following references with careful discussion regarding the mechanism of action of soy isoflavone genistein on thyroid (1,2) as well as of different nutraceuticals on cadmium-induced thyroid toxicity (3):

1) Marini H, Polito F, Adamo EB, Bitto A, Squadrito F, Benvenga S. Update on genistein and thyroid: an overall message of safety. Front Endocrinol (Lausanne). 2012 Jul 31; 3:94. doi: 10.3389/fendo.2012.00094. PMID: 23060856; PMCID: PMC3459182.

2) Bitto A, Polito F, Atteritano M, Altavilla D, Mazzaferro S, Marini H, Adamo EB, D'Anna R, Granese R, Corrado F, Russo S, Minutoli L, Squadrito F. Genistein aglycone does not affect thyroid function: results from a three-year, randomized, double-blind, placebo-controlled trial. J Clin Endocrinol Metab. 2010 Jun;95(6):3067-72. doi: 10.1210/jc.2009-2779. Epub 2010 Mar 31. PMID: 20357174.

3) Benvenga S, Marini HR, Micali A, Freni J, Pallio G, Irrera N, Squadrito F, Altavilla D, Antonelli A, Ferrari SM, Fallahi P, Puzzolo D, Minutoli L. Protective Effects of Myo-Inositol and Selenium on Cadmium-Induced Thyroid Toxicity in Mice. Nutrients. 2020 Apr 26;12(5):1222. doi: 10.3390/nu12051222. PMID: 32357526; PMCID: PMC7282027.

  • Thank you for the great suggestions. We have incorporated them in the text to improve the paper.